# Suggested Reference Ranges of Blood Mg and Ca Level in Childbearing Women of China: Analysis of China Adult Chronic Disease and Nutrition Surveillance (2015)

**DOI:** 10.3390/nu13093287

**Published:** 2021-09-20

**Authors:** Huidi Zhang, Yang Cao, Pengkun Song, Qingqing Man, Deqian Mao, Yichun Hu, Lichen Yang

**Affiliations:** National Institute for Nutrition and Health, Chinese Center for Disease Control and Prevention, Key Laboratory of Trace Element Nutrition, National Health Commission of the People’s Republic of China, Beijing 100050, China; zhanghuidi1114@126.com (H.Z.); Yasmine0814@163.com (Y.C.); songpk@ninh.chinacdc.cn (P.S.); manqq@ninh.chinacdc.cn (Q.M.); maodq@ninh.chinacdc.cn (D.M.); huyc@ninh.chinacdc.cn (Y.H.)

**Keywords:** reference range, magnesium, calcium, Chinese childbearing women

## Abstract

Background: Magnesium and calcium play a variety of biological roles in body functions. Reference values of these elements have not yet been systematically determined in China, especially in childbearing women. We proposed to establish the reference range of Mg, Ca, and Ca/Mg ratio in plasma and whole blood for 18–44 years healthy childbearing women in China. Method: A total of 1921 women of childbearing age (18–44 years) were randomly selected from the 2015 China National Nutrition and Health Survey by taking into account the regional types and monitoring points. Among them, 182 healthy women were screened out with a series strict inclusion criteria to study the reference ranges of elements. Fundamental indicators (weight, height, waist, blood pressure, total cholesterol, triglyceride, high-density lipoprotein cholesterol, low-density lipoprotein, fast glucose, HbA1c, blood pressure, uric acid) and elements concentrations in plasma and whole blood were collected. The 2.5th to 97.5th was used to represent the reference range of Mg, Ca, and Ca/Mg ratio. Results: The reference range of Mg, Ca, and Ca/Mg ratio in plasma were 0.75–1.13 mmol/L, 2.27–3.43 mmol/L, and 2.41–3.44, respectively. Additionally, the reference range of Mg, Ca, and Ca/Mg ratio in whole blood were 1.28–1.83 mmol/L, 1.39–2.26 mmol/L, and 0.90–1.66, respectively. According to the established reference range, the prevalence of magnesium deficiency was 4.79% in 1921 childbearing women, 21.05% in type 2 diabetes, and 5.63% in prediabetes. Conclusion: The reference values of Mg, Ca and Ca/Mg proportion in plasma and entire blood of healthy childbearing women can be utilized as a pointer to assess the status of component lack and over-burden. The lower limit of plasma Mg is in good agreement with the recommended criteria for the determination of hypomagnesemia.

## 1. Introduction

Magnesium and calcium are both vital components which act in a facilitated way to preserve physiologic cellular and tissue capacities [1]. Magnesium is the second abundant intracellular cation and it is required for the structure function of protein, nucleic acids, and DNA and RNA synthesis [2,3]. Calcium is the fifth most abundant element in the body and it plays an central role in regulating transport systems of intestinal absorption, renal reabsorption, and bone turnover [4,5].

These two elements are vital in supplying strength to bones that support locomotion [6]. Additionally, they play a vital part within the prevention and treatment of numerous diseases, such as cardiovascular infections [7,8] and type 2 diabetes [9].

Much of the literature discusses the antagonism of Mg and Ca, which indicates that a magnesium deficit would attenuate calcium channel-blocking effect and then stimulate an inflammatory response [10,11,12]. Background calcium/magnesium ratio can affect studies of any mineral individually [13]. Therefore, it is very important to analyze the ratio of calcium to magnesium while evaluating the distribution of calcium and magnesium in the body.

Lacking are studies evaluating Mg, Ca, and Ca/Mg ratio distribution at the same time in a healthy population; such study could help inform the biomarker use in assessing its ability to improve a deficiency or an overload status of elements.

The knowledge of the reference ranges for these elements in China is not available, especially in childbearing women. The aim of this study was to establish reference values of magnesium, calcium, and calcium/magnesium ratio for women of reproductive age, 18–44 years, in a representative Chinese population.

## 2. Materials and Methods

### 2.1. Subjects

The study was based on the 2015 Chinese Adult Chronic Disease and Nutrition Surveillance, a nationally representative cross-sectional survey. A total of 1921 women of childbearing age were randomly selected from the whole population by taking into account the regional types and monitoring points. Then, we selected healthy childbearing women (18–44 years) with a series of criteria. The inclusion criteria contained the following indicators in normal range: BMI (18.5–24.0 kg/m^2^), total cholesterol (TC, <5.2 mmol/L), triglyceride (TG, 0.56–1.7 mmol/L), low-density lipoprotein (LDL, <3.12 mmol/L), high density lipoprotein cholesterol (HDL-C, 1.04–2.07 mmol/L), uric acid (UA, ≤360 μmol/L), systolic blood pressure (SBP, 90–140 mmHg), diastolic blood pressure (DBP, 60–89 mmHg), fasting glucose (FG, 3.9–6.1 mmol/L), hemoglobin A1c (HbA1c, 4–6%), Hb (115–150 g/L), and heart rate (60–100 t/min). Exclusion criteria included: (1) smoking or having a history of smoking and (2) drinking or having a history of alcohol consumption in the past 12 months. All subjects gave informed consent during the study. The study was conducted in accordance with the Declaration of Helsinki, and the protocol was approved by the Ethics Committee of the National Institute of Nutrition and Health, Chinese Center for Disease Control and Prevention.

### 2.2. Data Collection

Medical examinations are collected by trained medical personnel in accordance with standardized procedures. Weight and height were measured to the nearest 0.1 kg or cm, by a Seca 213 Portable Stadiometer Height-Rod and a Seca 877 electronic flat scale, respectively. Body mass index (BMI) was calculated as weight (kg) divided by height in square meters (m^2^). The waist circumference was measured with a tape measure with an accuracy of 0.1 cm. SBP (mmHg) and DBP (mmHg) were evaluated using an Omron HBP-1300 professional BP monitor. Venous blood was collected and divided into anticoagulant tube and serum separation tube. The blood samples in the serum separation tube were centrifuged at 3000× *g* immediately after blood collection for 15 min, 20–30 min after the blood was taken and then divided into serum aliquots and frozen at −80 °C for subsequent assays. 

Serum fasting glucose, HDL-C, LDL, TC, TG, and UA were measured enzymically using an automatic biochemical analyzer (Hitachi 7600, Tokyo, Japan). Hb was determined by cyanide and high iron method, and HbA1c was determined by HPLC (Waters E2695, Milford, MA, USA). Element concentrations were measured by inductively coupled plasma mass spectrometry (ICP-MS, PerkinElmer, NexION 350, Waltham, MA, USA).

### 2.3. Determination of Mg and Ca in the Plasma and Whole Blood

Plasma element concentration was measured by 0.5% (*v*/*v*) high-purity nitric acid dilution (1:20), while whole blood element was measured by 0.5% (*v*/*v*) high-purity nitric acid and 0.05% (*v*/*v*) Triton X-100 dilution (1:25). The precision and accuracy of the analysis were monitored at 10-sample intervals using the quality control samples (Clinchek Level-2, Munich, Germany; Seronorm, Level-2, Billingstad, Norway). Since both magnesium and calcium are in ug/L, we first needed to convert them to mmol/L. The calcium/magnesium mole ratio was calculated by calcium concentration and magnesium concentration. The coefficient of variation between and within batches of Mg was 2.33% and 1.19%, respectively. The coefficient of variation of Ca between and within batches was 1.23% and 2.62%, respectively. The recovery of Mg and Ca were 93.44% and 97.63%, respectively. 

### 2.4. Statistic Analysis

SPSS 19.0 was used for statistical analysis. The results for descriptive characteristics were expressed as geometric mean (GM), median, P2.5, and P97.5. The International Union of Clinical Chemistry (IFCC) recommends at least 120 observations to estimate reliable reference values [14]. They defined reference intervals for clinical biomarkers based on concentration estimates of 2.5 and 97.5% of clinical biomarkers in the reference population. The differences between the two groups were tested by Student t test. The differences between more than two groups were detected by one way ANOVA.

## 3. Results

### 3.1. Basic Characteristics

After excluding the participants who had missing data or who failed to meet the inclusion criteria, the remaining 181 healthy childbearing women were included in this study. The basic characteristics, blood lipid, blood pressure, blood glucose, and other information of this group, are collected and summarized in Table 1. The mean age of the study participants was 27.98 years. Additionally, all the indexes were in the normal range.

The plasma concentrations of Mg, Ca, and Ca/Mg in the study population are shown in Table 2. These values are stratified by age and area factors. The reference ranges of plasma magnesium and calcium are 0.75 mmol/L and 2.27–3.43 mmol/L, respectively. The calcium magnesium ratio ranges from 2.41 to 3.44. Within the three element groups, there were no significant differences in results when stratified by age, region, and place of residence.

### 3.2. Whole Blood Concentrations of Mg, Ca, and Ca/Mg in Healthy Childbearing Women

The concentrations of the population by age group and area factors of whole blood elements are presented in Table 3. The reference whole blood ranges of Mg, Ca, and Ca/Mg ratio for healthy childbearing women were 1.28–1.83 mmol/L, 1.39–2.26 mmol/L, and 0.90–1.66, respectively. Mg concentrations were significantly different among age groups (*p* = 0.030). Ca concentrations and Ca/Mg ratio were markedly differed in the comparison of different area group (*p* = 0.005, *p* = 0.001). 

### 3.3. Plasma Magnesium Reference Ranges in Various Countries and the Deficiency of Magnesium under Different Standards in Childbearing Women

Table 4 presents the reference ranges of plasma Mg in different countries. Additionally, Table 5 shows the deficiency rate of Mg in 1921 childbearing women from China Adult Chronic Disease and Nutrition Surveillance (2015) according to different cut-off value of different countries. In the 1921 childbearing women, there were 38 subjects with type 2 diabetes and 160 subjects with prediabetes. Thus, we also calculate the Mg deficiency rate in type 2 diabetes and prediabetes with different cut-off values.

## 4. Discussion

As Apostoli [20] indicated, the reference range for an element should be established periodically and separately, because they can differ from gender to gender and be affected by dissimilar environmental scenarios. In this study, we finally enrolled 182 healthy individuals from a population of thousands of 18–44 years childbearing women to establish the reference range of Mg, Ca, and Ca/Mg ratio. All these data were nested on the latest national representative survey in China, which could give a reliable guidance to nutritional status assessment. We also detected the distribution stratified by age group and area factors. New reference ranges for Mg, Ca, and Ca/Mg were calculated by changing age group, area, and residence.

To date, many of the studies used to determine the current magnesium reference interval are based on magnesium distribution in the population rather than health outcomes [15,16]. In our research, we conducted the study with a series strict criteria to exclude the effects of some common chronic diseases, such as hypertension, hyperlipidemia, and abnormal blood glucose. Thus, the reference values can more purely reflect the natural distribution of elements in the human body.

The range from 2.5th to 97.5th are commonly used to indicate the reference interval [21]. In the current study, the overall reference range of 18–44 years healthy women for plasma Mg was 0.75–1.13 mmol/L, which was generally in the same range as that studied in 20–50 years healthy Iranian women (0.75–1.03 mmol/L). The initial study of reference range of magnesium (0.75–0.95 mmol/L) was first derived from NHANES I in 1974 [15], which was based on the distribution of serum magnesium in a normal population. Our upper limit of plasma Mg was higher than that assessed in America [15], Mexico [16], and Canada [18], and the lower limit was relatively same. Estimating the 2.5th concentration (0.75 mmol/L) as the cut-off value for identifying hypomagnesemia in the population, it is in good agreement with the previously recommended value [22]. Plasma Mg levels were not influenced by age, area, or residence in this study. We also detected the whole blood Mg concentrations, and the reference range was 1.28–1.83 mmol/L. The whole blood Mg were also not influenced by area and residence, but clearly the 26–35 years group had the highest Mg values than other age groups.

The reference range of Ca in plasma and whole blood was 2.27–3.43 mmol/L and 1.39–2.26 mmol/L, respectively. The normal range of serum Ca proposed by one study was 2.10–2.60 mmol/L [23]. A Ca concentration lower than 2.10 mmol/L is considered to have the possibility of hypocalcemia. In our finding, the lower cut-off was higher than 2.1 mmol/L, indicating that this value is suitable for evaluating Ca status for individuals. 

As it mentioned before [13], due to the competitive antagonistic effect of Ca and Mg, the Ca/Mg ratio can more comprehensively assess the nutritional status of elements in the body. The Ca/Mg ratio in healthy childbearing women was 2.41–3.44 and 0.90–1.66 in plasma and whole blood, respectively. Limited evidence to date, mainly in the form of cross-sectional studies, suggests that dietary calcium/magnesium intakes should be between 1.7 and 2.6 [24]. However, there is no reference data for the Ca/Mg ratio in the blood.

In this study, we not only screened the healthy people to establish the reference range of Mg, but also analyzed the prevalence of magnesium deficiency for this age group in the whole population according to the lower limit of the reference range. In the whole 1921 18–44 years subjects, the prevalence of Mg deficiency was 4.79% as the cut-off value of deficiency is 0.75 mmol/L. And when estimated among type 2 diabetes and prediabetes of the 1921 childbearing subjects, the Mg deficiency rate was 21.05 and 5.63%, respectively. We also evaluated Mg deficiency with the cut-off values used in Mexico [16] (0.71 mmol/L), Canada [18] (0.71 mmol/L) and Japan [19] (0.55 mmol/L). When 0.71 mmol/L was taken as the cut-off value, the prevalence of Mg deficiency in whole childbearing women, type 2 diabetes, and prediabetes was 1.56, 15.79 and 0.63%, respectively. However, there was no subject with a Mg concentration lower than the Japanese cut-off value 0.55 mmol/L. The analysis indicated that there may be great differences in the prevalence of Mg deficiency when estimating with different reference ranges. It is very important to establish the reference range of elements based on different national population.

The main advantage of this study is our efforts to establish reference ranges for elements in a variety of human mediums, including plasma and whole blood, which are most beneficial for mammalian enzyme activity and health. Second, it can provide a pure estimate of the reference value of the population of reproductive age 18–44 years, using high inclusion criteria and laboratory quality control.

One of the limitations of this study is that the dietary data of this population was not available in the research. Therefore, there are some weaknesses in assessing the effect of diet on the reference range. Secondly, considering the physiological mechanism, the indicators of inflammatory were not available in this paper, so there is no way to explore the effect of inflammation on element level. Thirdly, as our study concerned a relatively small number of cases, the results need to be validated in a larger population in the future.

In conclusion, this study provides reference ranges for magnesium, calcium, and calcium/magnesium in the plasma and whole blood of Chinese women of reproductive age 18–44 years. These data can be utilized to assess the clinical health and physical burden of the population. 

## Figures and Tables

**Table 1 nutrients-13-03287-t001:** Basic characteristics of the study population.

Variables (*N* = 182)	Median	GM	P25	P75
Age (years)	27.98	28.72	24.08	34.79
Height (cm)	157.3	157.02	153.3	161.5
Weight (kg)	51.8	51.95	48.58	55.63
BMI (kg/m^2^)	21.18	21.07	19.9	22.3
Waist (cm)	71.18	71.41	67.53	75.1
**Blood lipid**				
TC (mmol/L)	4.13	4.07	3.73	4.49
TG (mmol/L)	0.73	0.73	0.58	0.96
LDL (mmol/L)	2.34	2.26	2.01	2.66
HDL (mmol/L)	1.4	1.41	1.24	1.58
**Blood pressure**				
SBP (mmHg)	114.00	113.99	107.67	120.75
DBP (mmHg)	70.33	70.99	66.25	76.33
**Blood glucose**				
Glu (mmol/L)	4.87	4.85	4.60	5.14
HbA1c (%)	4.80	4.76	4.40	5.20
**Others**				
Hb (g/L)	137.91	135.68	127.88	144.27
UA (umol/L)	240.95	234.09	204.5	277.3
Heart Rate (t/min)	76.67	77.53	71.92	83.42
Vitamin D	16.29	15.68	12.10	21.60

BMI, Body Mass Index; TC, Total Cholesterol; TG, Triglycerides; LDL, Low-Density Lipoprotein; HDL-C, High Density Lipoprotein Cholesterol; UA: Uric Acid; SBP, Systolic Blood Pressure; DBP, Diastolic Blood Pressure; FG, Fasting Glucose; HbA1c, Hemoglobin A1c.3.1. Plasma Concentrations of Mg, Ca and Ca/Mg in Healthy Childbearing Women.

**Table 2 nutrients-13-03287-t002:** Plasma concentrations of Mg, Ca, and Ca/Mg in healthy childbearing women.

Variables	*N*	Mg (mmol/L)					Ca(mmol/L)					Ca/Mg				
		GM	P50	P2.5	P97.5	*p*	GM	P50	P2.5	P97.5	*p*	GM	P50	P2.5	P97.5	*p*
**Total**	182	0.90	0.90	0.75	1.13		2.63	2.58	2.27	3.43		2.93	2.93	2.41	3.44	
**Age group**						0.641					0.191					0.140
18–25 years	68	0.89	0.89	0.75	1.11		2.63	2.57	2.23	3.50		2.96	2.94	2.59	3.43	
26–35 years	70	0.90	0.90	0.71	1.23		2.66	2.59	2.30	3.68		2.95	2.92	2.38	3.58	
36–45 years	44	0.89	0.90	0.74	1.02		2.56	2.54	2.24	3.07		2.87	2.85	2.39	3.43	
**Area**						0.818					0.346					0.604
East	64	0.90	0.89	0.74	1.16		2.63	2.59	2.22	3.43		2.92	2.91	2.38	3.57	
Mid	92	0.89	0.90	0.73	1.17		2.64	2.58	2.27	3.61		2.95	2.94	2.45	3.47	
West	26	0.88	0.89	0.73	0.99		2.57	2.55	2.33	3.13		2.91	2.86	2.52	3.43	
**Residences**						0.726					0.692					0.323
City	79	0.90	0.90	0.77	1.17		2.61	2.57	2.27	3.24		2.91	2.91	2.44	3.43	
Rural area	103	0.89	0.89	0.72	1.13		2.63	2.58	2.25	3.52		2.95	2.96	2.39	3.50	

**Table 3 nutrients-13-03287-t003:** Whole blood concentrations of Mg, Ca and Ca/Mg in healthy childbearing women.

Variables	*N*	Mg (mmol/L)					Ca (mmol/L)					Ca/Mg				
		GM	P50	P2.5	P97.5	*p*	GM	P50	P2.5	P97.5	*p*	GM	P50	P2.5	P97.5	*p*
**Total**	182	1.50	1.49	1.28	1.83		1.77	1.75	1.39	2.26		1.18	1.16	0.90	1.66	
**Age group**						0.030					0.639					0.346
18–25 years	68	1.52	1.51	1.32	1.81		1.77	1.71	1.42	2.47		1.16	1.15	0.93	1.62	
26–35 years	70	1.48	1.48	1.24	1.94		1.79	1.76	1.41	2.26		1.21	1.20	0.93	1.70	
36–45 years	44	1.48	1.47	1.28	1.88		1.75	1.80	1.37	2.29		1.18	1.16	0.77	1.66	
**Area**						0.253					0.005					0.001
East	64	1.50	1.50	1.24	1.94		1.70	1.65	1.37	2.25		1.13	1.13	0.83	1.70	
Mid	92	1.48	1.47	1.29	1.81		1.83	1.87	1.43	2.41		1.24	1.22	1.00	1.66	
West	26	1.52	1.50	1.28	1.77		1.73	1.75	1.39	2.08		1.14	1.15	0.85	1.45	
**Residences**						0.252					0.367					0.951
City	79	1.51	1.48	1.30	1.88		1.79	1.80	1.37	2.46		1.19	1.16	0.89	1.67	
Rural area	103	1.48	1.49	1.26	1.79		1.75	1.73	1.39	2.23		1.18	1.17	0.89	1.67	

**Table 4 nutrients-13-03287-t004:** The reference ranges of plasma Mg in different countries.

Regions	*N*	Population	Reference Range of Plasma Mg (mmol/L)
China (this study)	182	18–44 years healthy women	0.75–1.13
America [15]	5083	18–44 years individual	0.75–0.95
Mexico [16]	3421	≥20 years women	0.71–0.92
Iran [17]	258	20–50 years healthy women	0.75–1.03
Canada [18]	511	20–39 years individuals	0.71–0.87
Japan [19]	31	26–89 years individuals	0.55–1.11

**Table 5 nutrients-13-03287-t005:** The prevalence (%) of Mg deficiency in 1921 childbearing women according to different standards.

Reference Standard	Cut-Off Value of Deficiency	Total	In Type 2 Diabetes	Prediabetes
China (this study)	0.75	4.79	21.05	5.63
America [15]				
Iran [17]				
Mexico [16]	0.71	1.56	15.79	0.63
Canada [18]				
Japan [19]	0.55	0.00	0.00	0.00

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
