# Peer review of "Suggested Reference Ranges of Blood Mg and Ca Level in Childbearing Women of China: Analysis of China Adult Chronic Disease and Nutrition Surveillance (2015)"

_nutrients, 2021, doi:10.3390/nu13093287_

Round 1

Reviewer 1 Report

All my comments have been taken into account.

Author Response

Thank you so much for your kind suggestion and help.

Reviewer 2 Report

In "subjects" section you mention " 1921 women of childbearing age were randomly selected from the whole population by taking into account the regional types and monitoring points." However in your abstract you mention "A total of 1921 18-44y childbearing women were enrolled from the 2015 China National Nutrition and Health Survey." Which of these two points is accurate? 

Moreover, you don't mention the age of gestation at which those women were enrolled in yourstudy. It is important to have a silid group of studyif it is going to be such a small number.

In your exclusion criteria you mention only smoking and drinking. Did you consider excluding healthy individuals who for any reason could be taking Mg or Ca oral supplements? Moreover, I understand that in order to identify "normal" individuals you need them to be "glabally healthy". However, a person may have ideal blood sugar level, blood pressure, TGs etc by taking proper medication. You don't mention oral medication use as an exclusion criterion.

How come and you did not include a control group of healthy non-pregnant women in order to identify possible differences between the two groups?

In discussion you have to point out that one of the limitations of your study is the relatively small number of cases studied (only 182). Therefore this study may only be used as hypothesis driving for future larger studies. Moreover since you do not mention the age of gestation of your subjects (not even the range) you have to make clear that such a normalization should be performed in a future study.

Author Response

Dear professor,

We very appreciate your careful reading of our manuscript and the valuable suggestions. We have carefully considered the comments and revised the manuscript accordingly. The comments can be summarized as follows:

Point 1: In "subjects" section you mention " 1921 women of childbearing age were randomly selected from the whole population by taking into account the regional types and monitoring points." However in your abstract you mention "A total of 1921 18-44y childbearing women were enrolled from the 2015 China National Nutrition and Health Survey." Which of these two points is accurate? 

Response 1: Thank you so much for the kind suggestion. Both of these expressions are used to describe the source of the research object. In order to ensure more accuracy, we replace the expression in the abstract with that in the article.

Point 2: Moreover, you don't mention the age of gestation at which those women were enrolled in your study. It is important to have a silid group of study if it is going to be such a small number. 

Response 2: Thank you so much for the kind suggestion. The study was hosted in 18-44y childbearing women. Due to the special physiological significance of this part of the population and from the perspective of the whole population, we give priority to the analysis of women of childbearing age aged 18-44. According to the principle of establishing the reference value range, the possible reference value can be calculated for more than 120 samples. However, in the follow-up study, we will continue to carry out continuous verification and correction in a larger sample population.

Point 3: In your exclusion criteria you mention only smoking and drinking. Did you consider excluding healthy individuals who for any reason could be taking Mg or Ca oral supplements? Moreover, I understand that in order to identify "normal" individuals you need them to be "glabally healthy". However, a person may have ideal blood sugar level, blood pressure, TGs etc by taking proper medication. You don't mention oral medication use as an exclusion criterion.

Response 3: Thank you so much for the kind suggestion. Due to the disease history, medication and dietary supplement records of this group, no data can be included for analysis at present. We can only analyze it in conjunction with its hematological indicators. It does introduce some confounding factors into the study. In the subsequent research, we will pay attention to the collection of this part of the content to ensure more accurate research.

Point 4: How come and you did not include a control group of healthy non-pregnant women in order to identify possible differences between the two groups?

Response 4: Thank you so much for the kind suggestion.The purpose of our study in this paper is to explore the normal reference range of this population, so we have not conducted a case control study on them for the time being. We will continue to conduct case-control studies in this population based on a larger sample size to explore the relationship between elements and different health outcomes.

Point 5: In discussion you have to point out that one of the limitations of your study is the relatively small number of cases studied (only 182). Therefore this study may only be used as hypothesis driving for future larger studies. Moreover since you do not mention the age of gestation of your subjects (not even the range) you have to make clear that such a normalization should be performed in a future study.

Response 5: Thank you so much for the kind suggestion.That's really the flaw of our study. We've fixed these content in the limitations section.

This manuscript is a resubmission of an earlier submission. The following is a list of the peer review reports and author responses from that submission.

Round 1

Reviewer 1 Report

The article fits into the Journal's scope. Subject matter is original and important. All research components are present and clearly stated. References are adequate. The text id adequately written.

The abstract is well‐written, concise and clear. Introduction is well‐written and concise. Objective of the study is clearly presented. Research methodology is given as citation – there is no methodology description what makes difficult for others to reproduce study by reading article. Results are logically presented. Minor repetition of data already included in tables, but rewriting can address this. Statistical significance of findings is given, however it is a bit difficult to follow the indicators of the data in the Tables. Conclusions are drawn from the analysis of the data collected. Statements and conclusions should be linked with goals, outcomes possible applications could be emphasized.  References are adequate and based on relevant literature. Some typographical errors can be easily fixed.

The main "weaknes" of this good article is lack of verification of data accuracy - Standard Reference Material analysis was performed, as the best method of determinations accuracy verification, , but precision and accuracy  of the results is not discussed here.

Author Response

Dear professor,

We very appreciate your careful reading of our manuscript and the valuable suggestions. We have carefully considered the comments and revised the manuscript accordingly. The comments can be summarized as follows:

Point : The main "weakness" of this good article is lack of verification of data accuracy - Standard Reference Material analysis was performed, as the best method of determinations accuracy verification, but precision and accuracy of the results is not discussed here.

Response: Thank you so much for the kind suggestion. It had been modified in the part of “2.3. Determination of Mg and Ca in the Plasma and Whole Blood”. We added the recovery of these two metals (Ca 97.63%, Mg 93.44%), which were calculated by standard reference material. The general acceptable range of recovery is 80-120%. According to this standard, the detection accuracy of two elements in this study is good.

Reviewer 2 Report

The work seems good and reasonable considering the desire to obtain normative data for reference.

We recommend publication.

Please comment more about these issues.

  1. How are they detecting the metal ions? Please show a cartoon.
  2. The ages 18-44 are stipulated based on the volunteer pool, not the biological ability to produce offspring. Please comment on this.
  3. Mention a few possible errors that might be involved with the analysis. Systematic errors etc.
  4. Mention a couple of papers from Daniella Buccela's research team regarding Mg homeostasis.

Author Response

Dear professor,

We very appreciate your careful reading of our manuscript and the valuable suggestions. We have carefully considered the comments and revised the manuscript accordingly. The comments can be summarized as follows:

Point 1:How are they detecting the metal ions? Please show a cartoon.

Response 1: Thank you so much for the kind suggestion. I’m sorry we did not detect metal ions in the study, but used ICP-MS to detect the total magnesium and calcium content in serum. We hope that with this high-throughput method, we can quickly detect the levels of elements in blood.

Point 2: The ages 18-44 are stipulated based on the volunteer pool, not the biological ability to produce offspring. Please comment on this.

Response 2: Thank you so much for the kind suggestion. Because in real life, even after China's two child policy is released, the real women of childbearing age are mostly concentrated in 18-44 years old. Therefore, this paper focuses on the research and exploration of this age group. We will verify the population not covered in this paper in the future.

Point 3: Mention a few possible errors that might be involved with the analysis. Systematic errors etc.

Response 3: Thank you for the suggestion. We have modified the limitation part in the article. Considering the physiological mechanism, the indicators of inflammatory were not available in this paper, so there is no way to explore the effect of inflammation on element level. Only from the perspective of health outcomes, we can get the reference range of normal population.

Point 4:Mention a couple of papers from Daniella Buccela's research team regarding Mg homeostasis.

Response 4: Thank you so much for your kind suggestion. We have read some of the research of professor Daniella Buccela. Research in the Buccella laboratory focuses on the study of metal ions and metalloenzymes in vitro and in vivo, with the ultimate goal of garnering a better understanding of the mechanisms that regulate metal homeostasis at the cellular level and of the molecular basis for the connection between disrupted metal balance and disease. This really give us a lot of inspiration and help. The majority of our research are mainly focus on population health, and the research on relative mechanism is relatively lacking. In the future, we will conduct more in-depth discussion on the mechanism research to better understand and explore the relationship between elements and human health.
